# Negotiating climate change: Science, policy, and the invisible power embedded in public discourse in Chinese social media

Kaijiao Zhang[1]*, Bingkang Qu[2], Qizhi Huang[3]

1 School of Literature and Media, Nanfang College Guangzhou, Guangzhou, Guangdong, China, 2 AI Lab, Ipsos (China) Consulting Co., Ltd., Guangzhou, Guangdong, China, 3 Faculty of Arts and Sciences, Beijing Normal University - Zhuhai Campus, Zhuhai, Guangdong, China

* evelynzhang2024@gmail.com

## Abstract

The use of social media has enabled diverse actors to engage in climate discourse, shifting climate change from a purely scientific issue to a broader global risk topic. Drawing on a decade of data from the Chinese platform Zhihu, this study employs social network analysis and correspondence analysis to examine the structural characteristics and phased evolution of social representations of climate change. The findings indicate that public discussions consistently anchor climate change in scientific evidence, forming a relatively stable core system. However, its peripheral structure has undergone continuous adjustment and expansion through the interaction between policy contexts and public communication, shifting from an emphasis on scientific consensus toward risk governance and economic pathways. This structural configuration reflects the dialogical and polyphasic nature of knowledge, while the economic perspective has emerged as a dominant representation of climate change in the Chinese context. This demonstrates the ongoing restructuring of social representations of climate change under the influence of policy and ideology, which may have significant implications for the public's perception of risk.

## 1. Introduction

While the world is becoming aware of the large-scale impacts of the climate crisis, geopolitical tensions and the risk of global fragmentation are undermining our risk perceptions and our ability to respond. Public understanding of science cannot be detached from broader social processes. However, a persistent gap exists between scientific and public perceptions of climate change, largely due to deep-seated political and cultural forces that hinder transformative environmental behavior [1]. The media has been recognized as a key agent in shaping climate consensus [2]. As a growing number of stakeholders, from scientists to policymakers, turn to social media for communication, and as the public increasingly uses these platforms to engage

**Data availability statement:** All data are available at the OSF: https://osf.io/amnv2/overview.

**Funding:** No funding was received. The funder provided support in the form of salaries for authors "BQ" but did not have any additional role in the study design, data collection and analysis, decision to publish, or preparation of the manuscript. The specific roles of these authors are articulated in the "author contributions" section.

**Competing interests:** The authors have declared that no competing interests exist.

with risk information, climate discourse on social media has emerged as a critical area of inquiry. These platforms, built on user-generated content, constitute an informal yet influential media environment. In this environment, multiple subjects actively participate in knowledge production [3]. In discussions on topics such as climate change, the amount of content, concentration, or other social cues on the platform can influence the public's perceived social consensus [4], which, together, shape the social meaning of climate change. This consensus is not a mere statement of fact but an expression of culture, values, worldviews, and political persuasions [4], and existing consensus may change as new discoveries are made, or public consensus may be challenged by the dissemination of offensive information [5]. Moreover, it has been suggested that the influence of social media in shaping people's risk awareness and health behaviors is more prominent in the chronic phase of a crisis than in the outbreak phase of a crisis event [6].

In China, influenced by both media technologies and the country's media governance system, climate change discourses have been largely dominated by governmental agencies and news institutions, following a top-down communication model. Recently, however, these discussions have increasingly migrated to digital social media platforms in more implicit and dispersed forms [7]. Social media platforms have become important alternative sources of information [8], exerting growing influence on climate change communication in the Chinese context. Accordingly, this study examines the content characteristics of climate change discourse on Chinese social media platforms and how such discourse interacts with broader sociocultural and political contexts. It can enhance our understanding of how the Chinese public perceives climate change. Moreover, it contributes to a deeper exploration of discursive interaction mechanisms within the science communication process and their broader societal implications.

## 2. Literature review and theoretical framework

### 2.1. Social representations theory and the climate change risk

Social Representations Theory (SRT) examines how social groups construct shared knowledge and meaning through communication [9]. It provides a framework for understanding how common-sense knowledge is produced and how this knowledge influences people's behavior and communication. The understanding developed by the public in response to complex topics or abstract topics can be referred to as social representations, a system of values, ideas, metaphors, beliefs, and practices. Structurally, social representations consist of a central system and a peripheral system. The central system is relatively stable and carries the group's core values and collective identity, and these elements tend to remain enduring in the absence of major social change or conflict. The peripheral system is more flexible and closely connected to everyday experience and situated practices, allowing it to adapt more readily to emerging social contexts, controversies, or new scientific developments [10]. Social representations are dynamic and competitive, embodying dynamic elements of knowledge, dependent on social conflict and controversy for their origins, and have a history of refinement and change over time [11]. In the practice of

climate change research, SRT emphasizes that the public does not simply receive information from scientists but actively reinterprets and reconstructs the meanings of climate change in public communications, media environments, and social interactions. Social representations should not be viewed as logical and coherent models or patterns of thinking. Instead, they represent a constantly evolving field of knowledge, where the way in which knowledge is generated and informs people's decision-making processes is relevant to the issue of climate change, among others, and where the common thinking of scientific thinking and societal consensus may be in conflict [12], revealing how societies dynamically form, maintain, or challenge cognitive structures and systems of meaning on particular topics.

According to previous studies, social representations can be conceptualized as user-generated semantic maps [13], which can help reveal implicit discourses or frames. This is because language and social practices are the primary ways in which social representations are formed and disseminated, thereby influencing people's opinions, attitudes, and related policies [14]. Conversely, shared social representations also affect how people use language, including the vocabulary and expressions they choose [15].

Accurately understanding the degree of climate change risk is critical in gaining public support for climate policies. However, public perceptions of the issue remain marked by significant divergences and controversy. For example, in the United States, environmental attitudes are closely tied to political affiliation, resulting in a pattern of symmetrical polarization, with Democrats (higher) and Republicans (lower) holding attitudes equidistant from the median. Pluralistic ignorance about climate issues and related policies was found in samples from China and Australia, where most people believe in human-caused climate change but incorrectly assume that their fellow citizens accept it [16]. Scholars argue that this consensus gap between the public and scientists is not simply due to a lack of knowledge or information but is closely related to individual psychological factors [17], as well as political ideology and sociodemographic backgrounds [18].

Further research indicates that this disagreement is also, in part, related to the linguistic features of risk communication. Psychological mechanisms, such as mental models, cognitive heuristics, and risk imagery, are constantly influenced by media reports, peer interactions, and communication channels, thereby modifying or reinforcing risk perceptions [19]. The public tends to interpret scientific information selectively within the framework of their existing knowledge and ideological orientations, such that scientific knowledge acquires additional meanings as it circulates in communication. Therefore, SRT provides a suitable framework for analyzing the public's understanding of climate change, as it helps delve deeply into how the public perceives it.

## 2.2. Social media and semantic networks

Traditional media, as well as all forms of online media, including social media, are regarded as informal resources for learning about complex political and scientific issues within the public sphere. Previous research reported on the potential of conducting science and research via social media, arguing that it serves not only the dissemination of entertainment-oriented content but is equally important for addressing serious topics [20]. Social media offers a more decentralized platform for discussion, enabling ordinary users to participate in public discussion and even challenge mainstream media narratives. Such communication practices effectively disrupt the traditional relationship in which scientists serve as authoritative producers of knowledge and laypeople as passive recipients, promoting the renegotiation of scientific knowledge within the public sphere [14]. However, some studies have shown that discussions on social media are not entirely open public dialogues but are often driven by influencers and organized networks. These actors shape public cognition through strategic communication practices, agenda setting, and public opinion management. Users face the challenge of processing diverse and often conflicting information, as even unbiased individuals are not entirely immune to the influence of misinformation [21]. This challenge is further compounded by the emergence of information cocoons created by algorithmic logic, AI applications, and commercial business models [22].

Social media posts have been described as proxies for broader public discourse [23] and as arenas in which different systems of meaning intersect and compete. Discussions of climate change in this space are not determined solely by

scientific facts but are also shaped by economic interests, political stances, and communication strategies. In other words, how people understand climate change depends on their social context and the interpretive resources available to them. Climate communication thus constitutes, to some extent, a process of negotiating meanings about the future [24]. Accordingly, prior research has traced the evolution of representations by analyzing digitally generated texts and time-series evidence [25]. In the Chinese context, diverse climate representations have emerged on platforms such as Weibo and Zhihu, where different interest groups compete for authority to define and interpret the issue, thereby shaping the range of acceptable policies and solutions [26]. Although media representations are not equivalent to cultural attitudes, beliefs, or values themselves, they provide important measurable indicators for observing domestic controversies and transnational ideological shifts [27]. The strategic deployment of linguistic choices and rhetorical structures (namely, how something is articulated) can be as revealing as the substantive content itself (that is, what is being said) in exposing underlying psychological processes [28]. Zaman found that people tended to describe the "disaster" brought by climate change as "an act of God," thereby adopting attitudes such as "let's see what happens" or "nothing doing," while endorsing "migration" as the primary response to climate change [29]. This underscores the power of language in constructing ideology. Therefore, by tracing the emergence and contestation of representations on social media, we can more clearly understand how climate risk is constructed, maintained, or challenged within the public sphere.

This study represents an exploratory exercise at the intersection of methodology and theory. There remains a notable lack of quantitative research examining the social representations of climate change in China. This study seeks to identify the types, structural configurations, and developmental trajectories of these representations by analyzing linguistic patterns and their temporal variation on social media platforms. The following research questions are proposed: RQ1: What constitutes the core social representations of climate change on social media? RQ2: How do linguistic and semantic connections in social media representations influence the public's understanding of climate risks? RQ3: How do major national policies or significant events influence social representations of climate change?

## 3. Research objects and methods

### 3.1. Research methods

In this study, these terms and words are used to reveal variations in public perception, areas of concern, and points of contention regarding climate change. Therefore, we used a mixed-method approach combining semantic network analysis and correspondence analysis to corroborate the research questions. Semantic network analysis can be regarded as an application of social network analysis in the technical domain, focusing on the logic of co-occurrence among linguistic units to reveal the text's ideational, rhetorical, and intellectual structure [30]. Correspondence analysis aims to illustrate important relationships between variable response categories through graphical maps [31].

### 3.2. Data collection and preprocessing

Zhihu's posters were used as the source of data collection. As the Chinese version of Quora, Zhihu optimizes content quality through user voting and algorithmic sorting, ensuring that high-quality answers receive greater exposure. Some Chinese scholars believe that interaction and dialogue between scientists and the public during scientific discussion have been effectively realized on Zhihu [32].

Data was collected using Python, with the keyword "climate" used to retrieve entries tagged accordingly on Zhihu. The dataset includes questions, answers, and articles associated with this tag, forming the basis of the study's corpus. The data collection period spanned from January 1, 2014, to December 31, 2024. To reduce noise and dimensionality while enhancing the accuracy of subsequent analyses, we applied a series of preprocessing steps to the title and abstract fields of posts and responses. These steps included deduplication, removing irrelevant content, and standardizing text formats. After cleaning, a total of M = 14,165 valid entries were retained as the final analytical sample. Finally, the sample was divided into three datasets based on the timing of two important events: China's participation in the Paris Agreement

negotiations and the formalization of the goal of carbon peaking and carbon neutrality. Building on this temporal framework, the three resulting datasets were categorized into Period 1, Period 2, and Period 3. All data were publicly accessible on the Zhihu platform, and the data collection and analysis methods complied with its terms and conditions.

## 4. Data analysis

Data processing was conducted using Python. First, the jieba module was used for Chinese word and phrase segmentation, filtering out meaningless demonstrative pronouns, conjunctions, adverbs of degree, and stop words. In the semantic network analysis part, Python is used to extract high-frequency keywords from the corpus, and the co-occurrence frequency of word pairs is calculated based on the co-occurrence relationships of keywords within the same text, which serves as the weighted value of the network edges. Subsequently, the node and edge data are imported into Gephi (version 0.10.1) to construct the word co-occurrence network and analyze the network's structural features and semantic connections among keywords. For the correspondence analysis, we used a word-count approach to compute term frequencies. However, given substantial variation in sample sizes across time periods, raw frequency counts were normalized to relative frequencies to ensure comparability. Afterward, we used SPSS (version 29) and Python to perform the corresponding analysis and its visualization.

## 5. Result

The study integrates semantic clustering and correspondence analysis to examine the construction and deployment of general themes and linguistic framings within specific social contexts. These analyses identify the semantic organization, structural configuration, and stage-based evolution of social representations of climate change.

### 5.1. The central structure of social representations of climate change

To explore how social representations of climate change reflect individual risk perceptions, we analyzed the distribution and co-occurrence patterns of high-frequency terms to uncover the associative structures and meanings underlying climate change discussions. Keywords were extracted from each corpus, and their co-occurrence frequency in the same text was counted. Each node represents a word, and the larger the node, the more frequently it is directly connected to other words, indicating a higher level of activity for that word within the network. A co-occurrence matrix is used to construct a list of edges between words, and the weight of each edge is the corresponding co-occurrence frequency. The thicker the edge, the higher the co-occurrence frequency. Node colors are generated based on the "modularity" algorithm and represent semantic communities. In Fig 1-3, we visualize the structure of the word co-occurrence network for three periods with the top 200 largest keywords.

Clusters of highly associated words are clearly visible in the semantic social network graphs. In the network visualization of Period 1 (Fig 1), the structure appears fragmented, with many modular clusters, indicating the emergence of relatively diverse discourse. In the green semantic cluster, keywords such as "temperature," "atmosphere," "precipitation," "ocean," "sun," and "elevated" form a typical Earth system causal chain. Rising temperatures trigger changes in atmospheric and oceanic circulation, which in turn affect precipitation and seasonal patterns, demonstrating a complex, coupled logic of natural scientific reasoning. This structure reflects a scientific mechanism framework and reinforces a cognitive schema that frames climate change as an irreversible systemic risk. The orange and purple semantic clusters revolve around terms such as "precipitation," "period," "history," "influence," and "mechanism," reflecting the users' attention to the temporal and regional mechanisms of climate change process in the discussion, showing a certain degree of public scientific literacy.

In the network graph for Period 2 (Fig 2), the most central nodes are concentrated in the orange cluster, extending the core set of scientific discourse from Period 1. The green cluster focuses on words such as "weather," "life," "rain," "air conditioner," and "recommend," indicating a growing tendency for users to link climate change with everyday life experiences.

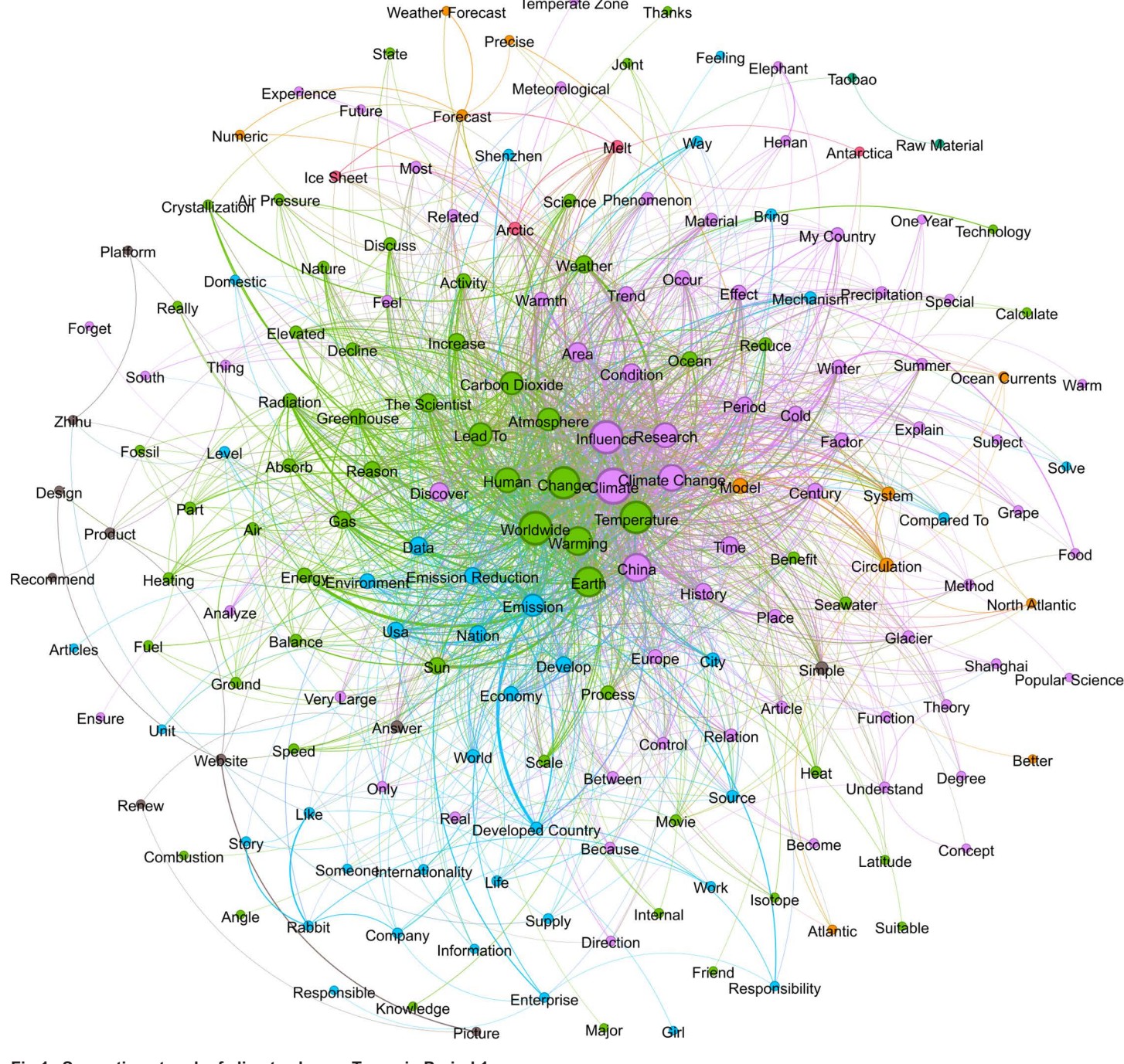

**Fig 1. Semantic network of climate change Terms in Period 1..**

The reddish-purple cluster, comprising words like "drought," "extinction," "sea level," "animal," "polar bear," and "Russia" form a relatively independent discourse system of catastrophic consequences, which extends peripherally from the core representation. These terms reflect the rise of disaster narratives and emotionally driven discourses. Notably, "politics," "United Nations," "agreement," "policy," and "policy" also appear on the boundaries of the modules, suggesting that the

**Fig 2. Semantic network of climate change Terms in Period 2.**

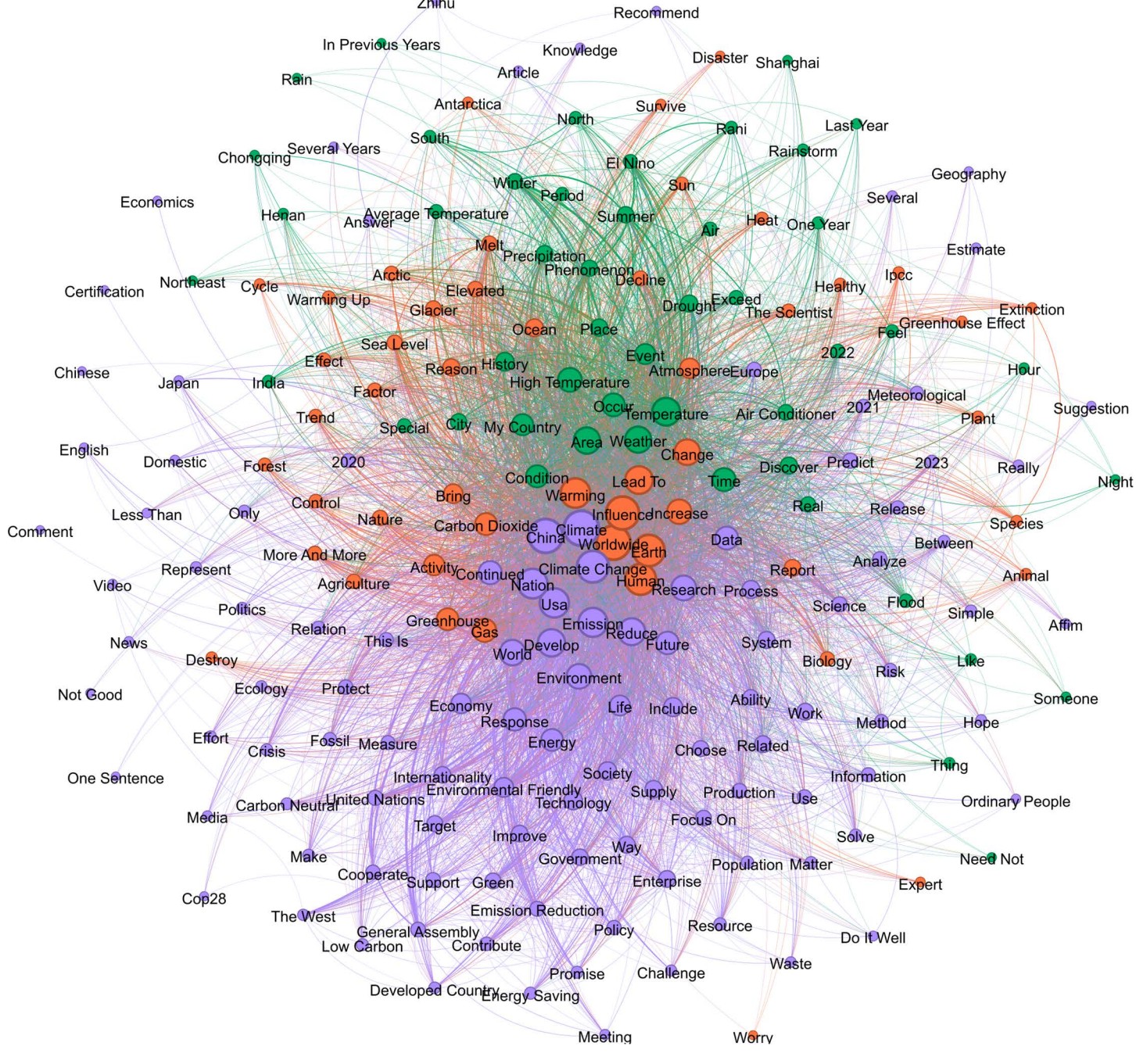

**Fig 3. Semantic network of climate change Terms in Period 3.**

context of international governance is gaining a place in the discussion and that geopolitical interventions in risk perception are beginning to be presented in the public, but not yet in a unified module.

The semantic network of Period 3 (Fig 3) is characterized by a more integrated structure and a convergence of discourse. The orange module inherits the logic of scientific consensus on the foundational perceptions of climate change

from the previous two periods but expands its boundaries to include terms such as "activity," "healthy," "animal," "IPCC," and "survive." This semantic expansion suggests that climate change is increasingly linking abstract climate mechanisms to living systems, human activity, and survival concerns. In the green cluster, terms such as "precipitation," "forecast," "radiation," "greenhouse," "temperature," "data," and "Meteo" are interlinked to show that the public is using weather and data models to understand the climate system. This module reinforces the scientific and technological cognitive pathway of rational modeling, which implies manageability or controllability policy logic. The purple community focuses on "policy," "developed country," "support," "enterprise," "responsibility," "low carbon," and "contribute," constituting a semantic system oriented towards institutional responsibility and the mobilization of governance. Unlike the perception of the phenomenon in the early stage, users at this stage pay more attention to actors and responsibility attribution, suggesting that the climate issue is gradually being reframed as an issue of institutional governance and legitimacy in the public's mind. This shift also reflects an increasing orientation toward practical mitigation strategies.

Taken together, the semantic networks across these three periods reveal a relatively stable scientific context, with these terms consistently anchoring discussions of the scientific consensus on climate change. At the same time, associations related to governance, responsibility, and solutions have grown increasingly prominent. Overall, the representations of climate change are not randomly distributed but are organized into identifiable semantic structures that shape the collective framework through which the issue is discussed. In this regard, RQ1 and RQ2 are addressed.

## 5.2. Temporal variation in climate change representations

After completing a preliminary analysis of the semantic relationships among keywords in the dataset, this study further explores the potential relationship between the time variable and the distribution of keywords. Correspondence analysis provides a low-dimensional representation of the associations or dependencies between the rows and columns of a contingency table, making it a useful tool for uncovering underlying structures in the data [33]. The procedure begins by computing distances among row profiles and column profiles. The measure of association employed is the chi-square distance between categorical variables [34]. These distances are then visualized through a two-dimensional scatterplot that aims to preserve the relational geometry of the original data as faithfully as possible. Following the three datasets mentioned earlier, we ran the word frequencies for each period. The top 60 words with the highest frequencies were retained, and a contingency table was constructed (Table 1). The row variables were the top 60 high-frequency words, the column variables were the three time periods, and the cell values represented the raw frequency as well as the relative frequency of the occurrence of each word in the corresponding time.

The contingency table was submitted to SPSS for correspondence analysis, and Table 2 demonstrates the key statistical indicators for correspondence analysis. This study extracted two dimensions. The first dimension captures the dominant variation in the data structure, while the second dimension provides additional refinement and interpretive depth [35]. Dimension 1 has a singular value of 0.495 and an inertia of 0.245, accounting for 65.8% of the variance explained. Dimension 2 has a singular value of 0.357 and an inertia of 0.128, explaining the remaining 34.2%. The two dimensions explain 100% of the total variance together, indicating all variability in the data can be described by these two dimensions. The chi-square test result was $\chi^2$ (184) = 418.698, p < .001, indicating a high correspondence in word distribution between time periods [31]. This result explains the significant changes in word frequency patterns of climate topics across time.

The analysis results were visualized using Python, and the resulting two-dimensional biplot shows the associations between periods and keywords depicted in Fig 4. Keywords are represented as blue dots, and periods are represented as red squares. The origin represents a hypothetical average keyword, i.e., a word that has an average association with each period [36]. The distance of a keyword from the Origin measures the differentiation of the word across time periods; if a keyword is evenly distributed across all time periods, it will be close to the Origin. Conversely, a keyword located farther from the origin is more closely linked to a particular time, often reflecting event-driven discourse. Keywords located near the origin, such as "human," "change," "ocean," "Earth," "environment," "data," and "warming," indicate that these terms

**Table 1. Contingency table of periods (columns) and words (rows).**

| | Period 1 frequency, n (%) | Period 2 frequency, n (%) | Period 3 frequency, n (%) |
|---|---|---|---|
| China | 469 (1.46%) | 2068 (1.17%) | 7603 (1.79%) |
| Global | 321 (1.00%) | 2030 (1.15%) | 7379 (1.74%) |
| Climate Change | 343 (1.07%) | 1659 (0.94%) | 6959 (1.64%) |
| Influence | 477 (1.48%) | 2429 (1.37%) | 6691 (1.58%) |
| Earth | 462 (1.44%) | 2471 (1.40%) | 5422 (1.28%) |
| Human | 303 (0.94%) | 2498 (1.41%) | 5319 (1.25%) |
| United States | 192 (0.60%) | 1377 (0.78%) | 5205 (1.23%) |
| Country | 175 (0.54%) | 1090 (0.62%) | 4176 (0.98%) |
| Temperature | 455 (1.41%) | 1701 (0.96%) | 3926 (0.92%) |
| Region | 204 (0.63%) | 1437 (0.81%) | 3721 (0.88%) |
| Lead to | 224 (0.70%) | 1465 (0.83%) | 3563 (0.84%) |
| Data | 179 (0.56%) | 1298 (0.73%) | 3293 (0.78%) |
| Reduce | 0 (0.00%) | 613 (0.35%) | 3188 (0.75%) |
| Development | 157 (0.49%) | 1060 (0.60%) | 3160 (0.74%) |
| Change | 448 (1.39%) | 1838 (1.04%) | 3065 (0.72%) |
| World | 159 (0.49%) | 944 (0.53%) | 2949 (0.69%) |
| High temperature | 0 (0.00%) | 0 (0.00%) | 2910 (0.69%) |
| Research | 326 (1.01%) | 1624 (0.92%) | 2888 (0.68%) |
| Increase | 172 (0.53%) | 833 (0.47%) | 2568 (0.60%) |
| Time | 204 (0.63%) | 1492 (0.84%) | 2549 (0.60%) |
| Energy Source | 0 (0.00%) | 0 (0.00%) | 2493 (0.59%) |
| Environment | 138 (0.43%) | 1081 (0.61%) | 2414 (0.57%) |
| Carbon Dioxide | 281 (0.87%) | 1058 (0.60%) | 2394 (0.56%) |
| Occur | 163 (0.51%) | 1063 (0.60%) | 2378 (0.56%) |
| Weather | 174 (0.54%) | 1024 (0.58%) | 2316 (0.55%) |
| Global Warming | 204 (0.63%) | 1363 (0.77%) | 2211 (0.52%) |
| Environmental protection | 0 (0.00%) | 0 (0.00%) | 2116 (0.50%) |
| Cope | 0 (0.00%) | 0 (0.00%) | 2111 (0.50%) |
| Future | 0 (0.00%) | 621 (0.35%) | 2063 (0.49%) |
| Emission | 199 (0.62%) | 554 (0.31%) | 2046 (0.48%) |
| Life | 0 (0.00%) | 862 (0.49%) | 1997 (0.47%) |
| Atmosphere | 240 (0.75%) | 1002 (0.57%) | 1925 (0.45%) |
| Activity | 121 (0.38%) | 873 (0.49%) | 1920 (0.45%) |
| Warming | 188 (0.58%) | 969 (0.55%) | 1882 (0.44%) |
| System | 114 (0.35%) | 699 (0.39%) | 1865 (0.44%) |
| Economy | 135 (0.42%) | 721 (0.41%) | 1828 (0.43%) |
| Event | 0 (0.00%) | 627 (0.35%) | 1769 (0.42%) |
| Technology | 0 (0.00%) | 608 (0.34%) | 1749 (0.41%) |
| Persistent | 0 (0.00%) | 0 (0.00%) | 1697 (0.40%) |
| History | 194 (0.60%) | 856 (0.48%) | 1673 (0.39%) |
| Reason | 165 (0.51%) | 855 (0.48%) | 1644 (0.39%) |
| Carbon emission | 0 (0.00%) | 0 (0.00%) | 1567 (0.37%) |
| City | 115 (0.36%) | 746 (0.42%) | 1548 (0.36%) |
| Air conditioner | 0 (0.00%) | 0 (0.00%) | 1496 (0.35%) |
| Ocean | 141 (0.44%) | 849 (0.48%) | 1491 (0.35%) |

*(Continued)*

**Table 1.** (Continued)

|  | Period 1 frequency, n (%) | Period 2 frequency, n (%) | Period 3 frequency, n (%) |
|---|---|---|---|
| Model | 137 (0.43%) | 0 (0.00%) | 1487 (0.35%) |
| Society | 0 (0.00%) | 0 (0.00%) | 1451 (0.34%) |
| Europe | 121 (0.38%) | 0 (0.00%) | 1434 (0.34%) |
| Phenomena | 0 (0.00%) | 0 (0.00%) | 1427 (0.34%) |
| Choice | 0 (0.00%) | 741 (0.42%) | 1424 (0.34%) |
| Precipitation | 0 (0.00%) | 578 (0.33%) | 1410 (0.33%) |
| Goal | 0 (0.00%) | 0 (0.00%) | 1390 (0.33%) |
| Enterprise | 0 (0.00%) | 0 (0.00%) | 1366 (0.32%) |
| Work | 0 (0.00%) | 822 (0.46%) | 1366 (0.32%) |
| Glacier | 0 (0.00%) | 579 (0.33%) | 1341 (0.32%) |
| Drought | 0 (0.00%) | 0 (0.00%) | 1321 (0.31%) |
| international | 0 (0.00%) | 0 (0.00%) | 1317 (0.31%) |
| El Nino | 0 (0.00%) | 0 (0.00%) | 1285 (0.30%) |
| Nature | 0 (0.00%) | 716 (0.40%) | 1285 (0.30%) |
| Climate | 739 (2.30%) | 4069 (2.30%) | 10622 (2.50%) |
| Sun | 177 (0.55%) | 636 (0.36%) | 0 (0.00%) |
| AD | 199 (0.62%) | 0 (0.00%) | 0 (0.00%) |
| Arctic | 114 (0.35%) | 561 (0.32%) | 0 (0.00%) |
| Biology | 0 (0.00%) | 688 (0.39%) | 0 (0.00%) |
| Capability | 0 (0.00%) | 556 (0.31%) | 0 (0.00%) |
| Century | 157 (0.49%) | 0 (0.00%) | 0 (0.00%) |
| Cold | 165 (0.51%) | 0 (0.00%) | 0 (0.00%) |
| Company | 0 (0.00%) | 554 (0.31%) | 0 (0.00%) |
| Control | 97 (0.30%) | 0 (0.00%) | 0 (0.00%) |
| Decrease | 106 (0.33%) | 0 (0.00%) | 0 (0.00%) |
| Developed Countries | 105 (0.33%) | 0 (0.00%) | 0 (0.00%) |
| Developing country | 96 (0.30%) | 0 (0.00%) | 0 (0.00%) |
| Effect | 141 (0.44%) | 616 (0.35%) | 0 (0.00%) |
| Emission Reduction | 221 (0.69%) | 0 (0.00%) | 0 (0.00%) |
| Energy | 133 (0.41%) | 0 (0.00%) | 0 (0.00%) |
| Factor | 135 (0.42%) | 568 (0.32%) | 0 (0.00%) |
| Greenhouse Gases | 102 (0.32%) | 0 (0.00%) | 0 (0.00%) |
| Information | 97 (0.30%) | 0 (0.00%) | 0 (0.00%) |
| Method | 0 (0.00%) | 596 (0.34%) | 0 (0.00%) |
| Outcome | 0 (0.00%) | 566 (0.32%) | 0 (0.00%) |
| Period | 207 (0.64%) | 0 (0.00%) | 0 (0.00%) |
| Radiation | 123 (0.38%) | 0 (0.00%) | 0 (0.00%) |
| Relationship | 112 (0.35%) | 563 (0.32%) | 0 (0.00%) |
| Relevant | 0 (0.00%) | 623 (0.35%) | 0 (0.00%) |
| Science | 108 (0.34%) | 0 (0.00%) | 0 (0.00%) |
| Scientist | 184 (0.57%) | 587 (0.33%) | 0 (0.00%) |
| Skin | 0 (0.00%) | 789 (0.45%) | 0 (0.00%) |
| Study | 0 (0.00%) | 573 (0.32%) | 0 (0.00%) |
| Trend | 141 (0.44%) | 0 (0.00%) | 0 (0.00%) |
| Understand | 99 (0.31%) | 0 (0.00%) | 0 (0.00%) |

*(Continued)*

**Table 1.** (Continued)

|  | Period 1 frequency, n (%) | Period 2 frequency, n (%) | Period 3 frequency, n (%) |
|---|---|---|---|
| Warm | 137 (0.43%) | 0 (0.00%) | 0 (0.00%) |
| Winter | 107 (0.33%) | 672 (0.38%) | 0 (0.00%) |
| Way | 0 (0.00%) | 553 (0.31%) | 0 (0.00%) |

**Table 2.** Summary table of the correspondence analysis.

| Dimension | Singular Value | Inertia | Chi-Square | Sig. | Proportion of Inertia | Cumulative |
|---|---|---|---|---|---|---|
| 1 | 0.495 | 0.245 | N/A[a] | N/A | 0.658 | 0.658 |
| 2 | 0.357 | 0.128 | N/A | N/A | 0.342 | 1 |
| Total | N/A | 0.373 | 418.698 | <.001[b] | 1 | 1 |

[a]not applicable.

[b]184 degrees of freedom.

exhibit a high degree of universality and stability across all three periods. This suggests that these terms constitute a shared semantic baseline. Discussions consistently revolve around these concepts, framing climate change as an environmental phenomenon grounded in the natural sciences. In this sense, they may be interpreted as forming the relatively stable core of climate change representations within the context examined, thereby providing further support for RQ1.

Dots representing the three periods appear in three distinct quadrants, revealing differences in public concern about climate change over time. The degree of clustering between periods and words, the angle of points on a map to the origin, and the proximity of points within the same quadrant can serve as interpretive cues for understanding the relationships between row and column variables [37]. It is worth noting that the angle in the biplot refers to the angle formed at the origin between vectors connecting the origin to a row variable and a column variable, respectively. A smaller angle indicates a stronger association between the two variables, whereas a right angle (90 degrees) suggests no relationship. An angle approaching 180 degrees implies a negative association [35]. Based on the criteria, we identified the most representative keywords by visually inspecting the biplot and computing their coordinates, which can be considered as distinctive phrases that characterize each specific period in contrast to the others [38].

The words most strongly associated with period 1 are "century," "emission reduction," "greenhouse gas," "radiation," "control," "energy," "science," "developed countries," and "developing countries". These terms suggest that, during this period, discussions around climate change were primarily driven by scientific research and had not yet been widely extended into policy or business. Discussions focused on the impacts of greenhouse gases, historical data, and scientific modeling of climate change. At the same time, the distribution of global carbon responsibilities was already being discussed, possibly in the context of climate governance in the era of the Kyoto Protocol. During this period, climate change was understood as a scientific topic, which aligns with Zhihu's early positioning as a platform for science-based knowledge exchange. The most relevant words in Period 2 are "biology," "capability", "effect", "company", "method", "relevant", "work," and "choice", etc., suggesting that climate change discussions during this period were no longer limited to physical and atmospheric sciences, but began to focus on the impacts of climate change on biological ecosystems, and began to gradually shift from a scientific issue to an economic and social one. The emphasis on corporate responsibility, for example, may reflect the fact that corporations began participating in climate governance at that time (e.g., through carbon trading and corporate sustainability strategies). The impacts of climate change policies on the labor market also began to receive attention, such as the green economy and the demand for labor resulting from the development of the new energy sector. In summary, the discussion of climate change has shifted from scientific debate to social action, focusing on biological impacts, corporate responsibility, and economic behavior, indicating that it has become a broader social issue.

 

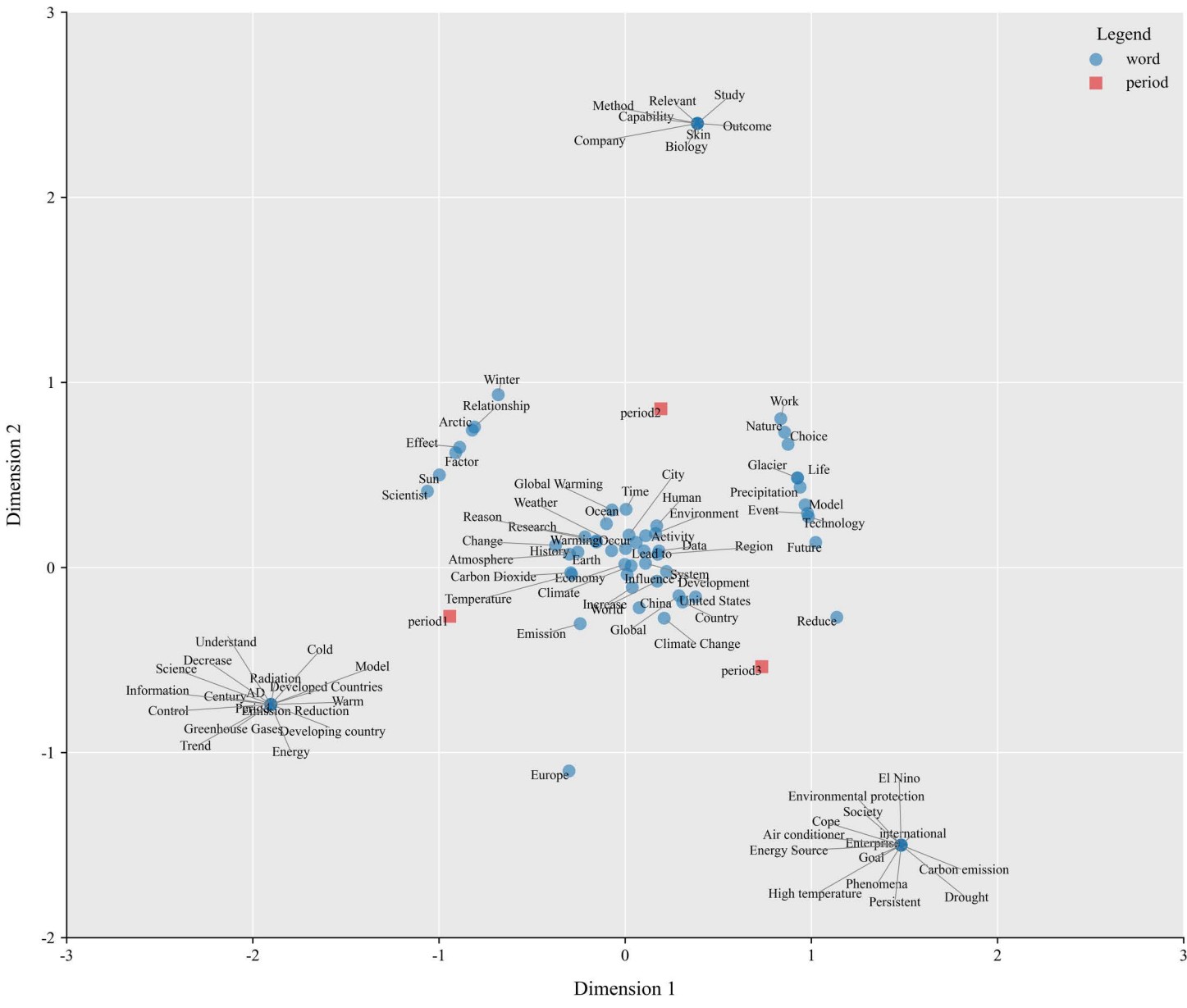

**Fig 4. Correspondence analysis map of periods and words.** Periods are red squares, words are blue circles.

As the scientific consensus expands, period 3 discourse becomes more pragmatic and urgent, focusing on the direct impacts of climate change, adaptation measures, energy transition, and international cooperation, reflecting the trend of climate change expanding into the realm of global policy and economics. The most relevant words during this period were "air conditioning," "carbon emissions," "business," "energy," "drought," "El Niño," "environmental protection," "high temperature," and "international." The focus of the discussion seems to have shifted to action on climate change, energy transition, technological development, and specific climate events (e.g., extreme heat, drought). This may imply a further shift in the focus of climate change discussions from scientific consensus and impact assessment to policy implementation and technological solutions. This transformation is, in fact, traceable. The visibility of climate change has increased in recent years due to the

growth of social media and the frequency of extreme weather [39]. To balance economic development with global climate responsibilities, the Chinese government proposed the goals of carbon peaking and carbon neutrality in 2015. This official initiative signaled that reducing carbon emissions and promoting energy transition had become central policy priorities in China. It is also widely recognized that global climate cooperation is more important than ever and that no country can tackle climate change alone. In sum, the spatial separation of period markers across different quadrants indicates that, although the representational structure associated with climate science remains identifiable, it has shifted noticeably in response to major events and policy developments in China. The social representations of climate change have moved from an early cognitive framework centered on scientific mechanisms toward a greater emphasis on economic practices and action pathways. These structural characteristics and their dynamic transformations provide an explanation for RQ3.

## 6. Discussion

### 6.1. The stability of the scientific narrative on climate change

Despite numerous social events occurring between 2014 and 2024 (e.g., the COVID-19 pandemic and fluctuating U.S.-China relations), climate change has consistently relied on scientific data and modeling to structure its interpretive framework. This has produced a stable and enduring scientific foundation, reflecting the persistence of the core elements of social representations [11]. This stability is evident in the prominence of high-frequency terms such as temperature, ocean, atmosphere, and Arctic, which repeatedly anchor discussions in measurable natural phenomena. As a result, climate change is primarily defined as a physical and environmental process grounded in scientific explanation. In addition, some Chinese participants equate climate change with smog or poor air quality, drawing on familiar environmental experiences to interpret an abstract global problem [7]. This reflects a process of objectification, whereby scientific knowledge is translated into concrete everyday phenomena. This cross-period stability stems from historical and cultural experiences and is relatively resistant to short-term external shocks. Given the long-term nature of climate change, whose impacts unfold over extended temporal scales, it is generally perceived as a phenomenon that is not fundamentally altered by short-term events. The second reason is the stability of China's climate policy. Despite internal economic pressures and changes in the external international environment, its overall policy direction has not undergone any fundamental shift. This policy continuity is not grounded in abstract or purely theoretical premises [40], but instead is based on scientific evidence and, to some extent, incorporates public feedback and lessons drawn from prior governance experience [40]. In addition, the complexity and global nature of climate change require that its discussion rely on scientific texts, such as the IPCC reports, the China Climate Change Blue Book, and the Bulletin of Ecological and Environmental Conditions in China, released by the Chinese Government. These scientific texts provide a standardized interpretive framework for the public. The emphasis on the stability of climate risk reflects its objectification in policy discourse, where it is treated as a tool for measuring and predicting risk rather than as an open, contestable system of knowledge.

### 6.2. Policy feedback and changes in the peripheral structure

Although there are stable focal points of interest, social representations of climate change on Zhihu exhibit a discernible pattern of dynamic change. Discussions have gradually shifted from explanations grounded primarily in climate science toward greater emphasis on impact assessment and solution pathways, with an increasingly prominent economic perspective linked to energy consumption. This suggests that the peripheral system of social representations is more context sensitive and adaptable, adjusting in response to new events and emerging information. In China, policy plays a significant role in shaping public perceptions of climate change [41]. As explained by policy feedback [42], policies possess structural power within the broader political ecosystem, which is influenced by existing public sentiment and, in turn, reshape the choices of various social actors. Early discussions focused on the causes of climate change and the scientific evidence, with the scientific community's consensus playing a central role. During the intermediate period, the cumulative effects of these themes became

increasingly evident, and representations gradually shifted from characterizing climate science to discussing the impacts of climate change and response strategies. This process can be viewed as a dynamic adjustment of peripheral characteristics triggered by various social contexts, such as the entry into force of the Paris Agreement, the increasing frequency of extreme weather events, and developments in China–U.S. climate diplomacy, which in turn continuously provided new contexts and interpretive frameworks for core elements. Meanwhile, reflecting the increasing role of market mechanisms in climate governance [43]. In recent years, the focus of discussion has shifted toward economic issues. This trend is closely linked to establishing carbon peaking and carbon neutrality targets, developing carbon markets, engaging climate finance actors, and advancing the sustainable development agenda [44]. In this process, policy does not function as a static, one-way transmission of directives, but rather as an ongoing social negotiation. It is reflected both in the implementation of policies that contribute to the reconstruction and renewal of climate representations and in the provision of cognitive and normative foundations that legitimize representations [45]. In other words, policy not only responds to public risk perceptions but also offers new anchoring points for understanding climate change, thereby enabling context-specific updates through the expansion of the peripheral structure. Consequently, communication practices surrounding these topics might thus need to adopt a more holistic perspective and language to connect to the public's representations of these issues.

## 6.3. Hegemonic representation and the economic perspective of climate change

Climate change is frequently employed to carry and communicate various ideological assumptions and projections [46]. For instance, preferences for climatic stability over change, or for framing climate change as a threat or an opportunity, reflect divergent ideological perspectives. As a result, it is inherently open to multiple, layered, and sometimes contradictory interpretations. Within the global media environment, climate change has come to be framed through a broadly shared representational framework, and our findings are consistent with this pattern. It implies that the public's perceptions of climate change share certain similarities, suggesting the existence of a cross-cultural representation of climate change [47,48]. However, it should be noted that differences in specific contexts and lived climate experiences may lead to variations in the content and organization of these representations, whose core feature lies in defining the mechanisms and boundaries through which power operates [49]. It is reflected in concrete practice in the dominance of representations and the marginalization of others. For example, analyses of nationally determined contributions submitted under the Paris Agreement suggest that environmental problems are often framed as solvable through technological improvement and efficiency gains, thereby reinforcing a liberal ecological framework [50].

A similar pattern can be observed in Chinese data. A dominant representation has emerged that frames economic instruments as primary solutions, emphasizing the macro-regulatory role of economic policy in the Chinese public sphere. In the co-occurrence networks and correspondence maps, enterprise-related terms frequently cluster with carbon emissions, energy governance, environmental regulation, and responsibility. Public discussions often cite IPCC reports, temperature thresholds, and carbon intensity indicators, linking emissions to economic growth. Within this framework, corporations are positioned as key actors in climate mitigation. Climate change is therefore understood less in terms of individual behavior and more as an issue of industrial transformation and policy-led governance. Framed within resource management and economic regulation, it becomes detached from moral obligation. It is as if climate change can be controlled simply by following the shortcuts of science-based economic governance. Such hegemonic representation defines what forms of discourse and action are considered legitimate [51]. It is difficult to reinterpret or contest because of its normative force. At the same time, it leaves space for emancipated representations to emerge through negotiation [52]. For example, some discussions frame climate change as an issue of responsibility between the Global North and South, offering a modification or refinement of the dominant representation. While some polemic representations (e.g., climate skepticism or climate conspiracy) are still at the socially contested and consultative stage, attempting to challenge existing dominant views and practices. These representations are often used as anti-science or anti-policy expressions, which may lead to the invisibility of uncertainty and the weakening of pluralistic knowledge.

Notably, situating climate issues within an economic framework and amplifying confrontational representations suggest that climate change is framed as a governable and manageable problem. This finding aligns with observations from semantic network analysis, which show that keywords with high centrality are primarily concentrated in rational discourses such as economics, technology, and science, with very few "catastrophic" or "apocalyptic" discourse nodes appearing. Such a pattern reflects a form of de-dramatizing discursive practice [53]. These linguistic strategies may attenuate public perceptions of urgency and irreversibility associated with climate change. This study reveals that the Chinese public's understanding of climate change has gradually solidified through interactions between policy contexts and public communication, affirming that climate risk is a socially constructed phenomenon [54].

## 7. Conclusion

This study draws on user-generated content from social networking platforms to examine climate change representations in China between 2014 and 2024. The findings indicate that climate change has consistently formed a stable core system grounded in natural scientific data within public discussions. However, its peripheral structure has evolved in response to climate policies, gradually shifting from an emphasis on scientific evidence toward risk governance and consolidating into a hegemonic representation that privileges economic instruments as the primary solution. This structural configuration acknowledges both the dialogic and polyphasic nature of knowledge, delineates legitimate problem definitions and solution pathways, and reflects the ongoing reconfiguration of social representations over time.

At the theoretical level, this study extends the application of social representations theory within digital media contexts. It demonstrates that climate change is interpreted through social lenses rather than treated as a purely scientific object. Instead, it is continuously objectified and reorganized within institutional environments and social interactions. At the practical level, this study helps to better understand the complex perceptions Chinese people hold regarding climate change, viewing it both as a hazardous climatic phenomenon and as an opportunity for economic transformation. However, the data reveal that institutional voices, benefiting from structural advantages in information dissemination, occupy dominant positions in the discursive field, thereby marginalizing grassroots perspectives and heterogeneous interpretations. Climate communication strategies could potentially enhance policy understanding and risk communication by connecting the physical world with our cultural imaginations, thereby appropriately expanding peripheral structures.

There are limitations to this study. Even though social media provides a platform for public discussion, there is still disagreement over how much influence it has in the policy process [27]. Therefore, it is difficult for us to adequately explain the prevalence of policies, or the eventual direction of policy choices based solely on what is available on social media. The study was also unable to distinguish the differential influence of various actors within the social media environment. In addition, the data acquisition process may be affected by platform algorithmic mechanisms, content review rules, and data interface limitations, leading to potential data bias or missing data that may limit the generalizability and generalizability of the findings. From a technical perspective, research methods may have limitations in handling complex semantics and polysemous terms, which can affect the accuracy of interpreting discussion content. Future research should adopt broader analytical frameworks rather than focusing solely on the scientific label "climate change" to more fully capture the complexity of common-sense representations. In addition, the role of different network actors in shaping the development of social representations warrants closer examination. Experimental approaches may further investigate how distinct representational structures influence public intentions to act.

## Author contributions

**Conceptualization:** Kaijiao Zhang.

**Data curation:** Bingkang Qu.

**Formal analysis:** Qizhi Huang.

**Methodology:** Kaijiao Zhang.

**Software:** Bingkang Qu.

**Supervision:** Kaijiao Zhang.

**Validation:** Kaijiao Zhang, Qizhi Huang.

**Writing – original draft:** Kaijiao Zhang.

**Writing – review & editing:** Bingkang Qu.

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
