## [Decision Letter · Decision Letter 0]

22 Feb 2026

PONE-D-25-65638Negotiating Climate Change: Science, Policy, and the Invisible Power Embedded in Public Discourse in Chinese Social MediaPLOS One

Dear Dr. Zhang,

Thank you for submitting your manuscript to PLOS ONE. After careful consideration, we feel that it has merit but does not fully meet PLOS ONE’s publication criteria as it currently stands. Therefore, we invite you to submit a revised version of the manuscript that addresses the points raised during the review process.

We look forward to receiving your revised manuscript.

Kind regards,

Gordon Sammut

Academic Editor

PLOS One

Journal Requirements:

2. In your Methods section, please include additional information about your dataset and ensure that you have included a statement specifying whether the collection and analysis method complied with the terms and conditions for the source of the data.

3. Please note that PLOS One has specific guidelines on code sharing for submissions in which author-generated code underpins the findings in the manuscript. In these cases, we expect all author-generated code to be made available without restrictions upon publication of the work. Please review our guidelines at https://journals.plos.org/plosone/s/materials-and-software-sharing#loc-sharing-code and ensure that your code is shared in a way that follows best practice and facilitates reproducibility and reuse.

4. We note that one or more of the authors are employed by a commercial company: “Ipsos (China) Consulting Co., Ltd. Beijing, China”

“The funder provided support in the form of salaries for authors “BQ” but did not have any additional role in the study design, data collection and analysis, decision to publish, or preparation of the manuscript. The specific roles of these authors are articulated in the ‘author contributions’ section.”

Additional Editor Comments:

In particular, I recommend addressing the suggestions to (a) restructure the discussion around the Research Questions, (b) addressing the social representations literature more directly, and (c) noting the pertinence of different communication genres like social media.

Reviewer's Responses to Questions

**Comments to the Author**

1. Is the manuscript technically sound, and do the data support the conclusions?

Reviewer #1: Yes

Reviewer #2: Partly

2. Has the statistical analysis been performed appropriately and rigorously? 

Reviewer #1: Yes

Reviewer #2: Yes

3. Have the authors made all data underlying the findings in their manuscript fully available?

Reviewer #1: Yes

Reviewer #2: Yes

4. Is the manuscript presented in an intelligible fashion and written in standard English?

Reviewer #1: Yes

Reviewer #2: Yes

5. Review Comments to the Author

Reviewer #1: The article addresses a topic of definite interest and, with adequate mastery of both the reference literature and research methods, positions itself at the intersection of the field of social representation theory (SRT) and studies on the communication of social problems (public discourse, digital media content analysis). The study is based on a rigorous and well-documented quantitative approach, with an interesting time frame (10 years) in order to understand the elements of climate change discourse in Chinese social media from a longitudinal perspective.

The results are significant.

The article is very clear and well written overall, except for a few oversights that need to be checked (see below).

The main limitation, which I would describe as structural and of which the author(s) seem quite aware (see Conclusions), is the basic assumption that all discourse on social media can be interpreted as public discourse. In addition to the issues already acknowledged by the author(s), there is the question of not considering, or consider little, social media – even attempting to control it from a technical or methodological point of view – as an arena in which highly heterogeneous actors operate (and therefore with different opportunities and structures of symbolic and discursive power). The focus solely on the linguistic and semantic dimensions of corpora tends to flatten the issue of 'who' produces discourse. Clearly, policy discourse reflects (as well as directly emanating from) government agencies, organised actors, NGOs, politics, but also fragmented actors: beyond the political and institutional characteristics of the Chinese context, in its specificity, discourse on social media is also developed by individual experts, influencers, celebrities, ordinary people as individuals, etc. I understand that this is not an easily solvable problem and that, above all, even if this focus could be added (to research already conducted), it is not certain that this would be efficient or capable of producing interesting inferences, but some form of identification and coding could have been included in the approach, as a variable (in addition to the period), of the actors who produce the discourse, even if only as a simple coding (type of actor, individual vs. collective, institution, etc.).

Or if this has been done, it is unclear (and has not been explored analytically).

I recommend reviewing certain parts of the discussion, where the connection with some of the key themes in the literature may appear somewhat forced and could be better articulated.

For some statements, which are probably plausible and in principle not falsifiable, or correct, the placement in the discussion of the results does not seem to be adequately supported – at least explicitly – by the data. Where, for example, is the data supporting this part? (p. 23):

For instance, a misalignment is often observed in public discussions about the attribution of responsibility, with the public tends to blame corporations for causing global warming while holding governments accountable for solving the problem; meanwhile, governments increasingly place the burden of action on corporate actors. This flow of responsibility discourse reflects an ongoing struggle over the fundamental question of “who is responsible.”

Some statements are not supported by references (bibliographical or primary data), at least not explicitly. Please correct. E.g. p. 24

For example, some residents in northern Chinese cities tend to…

Below are some basic suggestion that I believe may be useful during the minor revision phase.

p. 12 When mentioning the software used, I think it is always good practice to mention the version used: SPSS, version?

p. 17 Check the sentence: the sentence The terms “science,” “developed countries,” and “developing countries.” seems incomplete, is there a verb missing? Or, directly, without a full stop, "... suggest that…"

Attention, check typos or error. I have noted some here, but it means that careful reading is required to ensure that there are no others.

p. 10 Correspondence analysis (not Correspondent)

p. 10 Chinese (not Chiese)

Reviewer #2: Dear Authors,

This paper is about representations and discourses of climate change on Chinese social media. The paper makes an important contribution to the literature, but would benefit from more clarity in terms of the presentation of findings, and the discussion of findings (particularly in terms of theoretical contributions). Please find my feedback below:

1.) The abstract mentions action-oriented and governance-focused perspectives, but these are not present in the text. The authors are advised to show how these are relevant to the study or its findings.

2.) The literature review is sound, and culminates in three research questions. However, in presenting the results, the findings are not structured around these questions. Instead of sticking to analytical categories on their own (e.g., as relating to word count, correspondence analysis, etc.), the Results section should be structured around the research questions (whilst also bringing out the findings of the separate analyses). This would ensure that the writing directly addresses the RQs.

3.) The Discussion lacks theoretical depth. Social representations feature in the literature review, but not in the Discussion (only occasionally, when discussing hegemonic representations). It would be ideal if social representations feature (1) theoretically, and (2) in view of the research questions on Page 10: How were these answered? What contributions do the findings make? Which theoretical framework is supported by the findings, or opposed? etc.

4.) The research questions (page 10) are "What social representations of climate change are most salient in public discourse on social media?", "To what extent is public consensus on climate science influenced by major government initiatives or significant events?", and "How does media discourse on climate change interact with the social construction of climate risk within social media environments?" However, keywords shift from representations to discourse to scientific narratives, meaning that the paper is less coherent than it could be. Authors are advised to re-write parts of the paper (where relevant) to ensure alignment between approaches (e.g., predominantly social representations, or predominantly a discourse-based approach).

5.) Contributions of this work, practical implications, and directions for future research, need to be highlighted more (in Conclusion chapter)

6. PLOS authors have the option to publish the peer review history of their article (what does this mean?). If published, this will include your full peer review and any attached files.

Reviewer #1: No

Reviewer #2: No

---

## [Author Response · Author response to Decision Letter 1]

11 Apr 2026

We thank the editor and reviewers for their valuable comments and constructive suggestions. We have carefully revised the manuscript accordingly, and a detailed, point-by-point response to all comments is provided in the uploaded Response to Reviewers document.

---

## [Editor Report · Decision Letter 1]

21 Apr 2026

Negotiating Climate Change: Science, Policy, and the Invisible Power Embedded in Public Discourse in Chinese Social Media

PONE-D-25-65638R1

Dear Dr. Zhang,

We’re pleased to inform you that your manuscript has been judged scientifically suitable for publication and will be formally accepted for publication once it meets all outstanding technical requirements.

Kind regards,

Gordon Sammut

Academic Editor

PLOS One
---

## [Editor Report · Acceptance letter]

PONE-D-25-65638R1

PLOS One

Dear Dr. Zhang,

I'm pleased to inform you that your manuscript has been deemed suitable for publication in PLOS One. Congratulations! Your manuscript is now being handed over to our production team.

Kind regards,

on behalf of

Professor Gordon Sammut

Academic Editor

PLOS One